# Selective Arterial Embolization of Pseudoaneurysms and Arteriovenous Fistulas after Partial Nephrectomy: Safety, Efficacy, and Mid-Term Outcomes

**DOI:** 10.3390/biomedicines11071935

**Published:** 2023-07-07

**Authors:** Romaric Loffroy, Amin Mazit, Pierre-Olivier Comby, Nicolas Falvo, Claire Tinel, Olivier Chevallier

**Affiliations:** 1Department of Vascular and Interventional Radiology, Image-Guided Therapy Center, François-Mitterrand University Hospital, 14 Rue Paul Gaffarel, BP 77908, 21079 Dijon, France; amin.mazit@chu-dijon.fr (A.M.); nicolas.falvo@chu-dijon.fr (N.F.); olivier.chevallier@chu-dijon.fr (O.C.); 2ICMUB Laboratory, UMR CNRS 6302, University of Burgundy, 21078 Dijon, France; 3Department of Neuroradiology and Emergency Radiology, François-Mitterrand University Hospital, 14 Rue Paul Gaffarel, BP 77908, 21079 Dijon, France; pierre-olivier.comby@chu-dijon.fr; 4Department of Nephrology and Renal Transplantation, François-Mitterrand University Hospital, 14 Rue Paul Gaffarel, BP 77908, 21079 Dijon, France; claire.tinel@chu-dijon.fr

**Keywords:** kidney tumor, partial nephrectomy, bleeding, pseudoaneurysm, arteriovenous fistula, embolization, interventional radiology

## Abstract

The primary objective was to evaluate the clinical success rate after endovascular embolization of iatrogenic vascular lesions caused during partial nephrectomy. The secondary objective was to evaluate the technical success and to assess potential effects on renal function. We retrospectively included consecutive patients from our center who underwent selective embolization to treat iatrogenic renal arterial lesions induced during partial nephrectomy between June 2010 and June 2020. The technical and clinical success rates and renal outcomes were collected. We identified 25 patients with 47 pseudoaneurysms and nine arteriovenous fistulas. Among them, eight were treated by coils only, eight by liquid embolization agents only, and nine by both. The technical success rate was 96% after the first attempt and 100% after the second attempt. The median follow-up was 27.1 ± 24.3 months. Clinical success, defined as no need for further hemostatic surgery during follow-up, was also obtained in 96% and 100% of patients with one and two attempts, respectively. Renal function estimated by the modification of diet in renal disease equation did not change significantly despite a mean 13.8% ± 15.1% decrease in kidney functional volume estimated by angiography. No complications were attributable to the endovascular treatment. No significant difference was found across embolization agents; however, the subgroup sizes were small. Endovascular embolization is safe and effective for treating iatrogenic arterial lesions after partial nephrectomy: success rates are high, complications are infrequent, and renal function is maintained. Recommendations by interventional radiology societies are needed to standardize this treatment.

## 1. Introduction

The incidence of kidney cancer has risen recently, in large part due to the increased number and quality of diagnostic imaging studies, with a substantial proportion of tumors being incidental discoveries [1,2]. This improvement in diagnostic performance is associated with an increased number of tumors being discovered at an early stage, when nephron-sparing surgery is possible. Partial nephrectomy (PN) is the reference standard treatment for small tumors [3,4]. PN is associated with longer survival in patients who have low pathological tumor-node-metastasis stages, as well as with greater preservation of the nephron capital with less postoperative renal failure compared to radical nephrectomy (RN) [5,6,7,8,9].

A disadvantage inherent in PN compared to RN is the higher postoperative complication rate [10], including a higher frequency of iatrogenic arterial lesions such as false arterial aneurysms (FAAs) and arteriovenous fistulas (AVFs) [11,12,13]. Thus, the expanding use of PN to treat kidney cancer has resulted in an increased number of iatrogenic arterial complications, which must be identified and treated. Bleeding, manifesting as hematuria, anemia, or surgical wound bleeding, is the main symptom and requires emergency treatment in a minority of cases. New flank pain should also suggest an arterial injury. Finally, in some patients, the lesion is discovered upon routine follow-up imaging studies.

Endovascular management by selective arterial embolization (SAE), when available, is consensually preferred as the first-line treatment for iatrogenic arterial lesions after PN, as it allows the exclusion of FAAs while ensuring maximal parenchymal preservation [14,15]. In a review of 30 studies including 105 patients with FAAs after PN, SAE was the first-line treatment in 101 (96%) patients [11]. Surgical hemostasis was the first-line procedure in patients with hemodynamic instability (2%) and the second-line procedure in the event of failed embolization (2%), as these situations usually require RN to stop the bleeding.

Although their incidence is increasing, iatrogenic arterial complications after PN remain rare, and few studies with substantial sample sizes are available to accurately assess the efficacy of endovascular embolization, identify the optimal embolization material, and evaluate effects on kidney function and volume. Thus, a 2015 literature review identified fewer than 150 reported cases of FAA after PN [16].

With this in mind, we conducted a study in 25 patients admitted to the Dijon University Hospital in 2010–2020 for iatrogenic renal arterial lesions after PN. The aim of the study was to assess the technical and clinical effectiveness of SAE, its impact on renal function and volume, and the recurrence rate.

## 2. Materials and Methods

### 2.1. Study Design and Patients

We conducted a single-center retrospective study at the Dijon University Hospital over the 10-year period extending from 1 June 2010 to 1 June 2020. We included consecutive patients who received endovascular treatment for renal FAA, AVF, or both at our hospital. All study data were anonymized. In compliance with French law, ethics committee approval was not required for this retrospective study of anonymized data. The requirement for informed patient consent was waived for the analysis of anonymized data. The study was conducted in accordance with the principles of the Declaration of Helsinki. 

### 2.2. Data Collection 

The study data were extracted from urological surgery reports, hospital discharge summaries, laboratory databases, computed tomography (CT) and angiography images and reports, and interventional radiology images and reports filed at the Dijon University Hospital and at any other healthcare institutions where patients received follow-up.

Creatinine clearance calculated according to the standardized modified diet in renal disease (MDRD) equation was collected within 7 days before and after PN, within 7 days before and after SAE, and 6 to 12 months after SAE, with the earliest value being chosen when several values were available [17]. Chronic kidney disease (CKD) was classified according to National Kidney Foundation/Kidney Disease Outcomes Quality Initiative (NKF KDOQI) criteria [18]. Finally, the loss of kidney functional volume induced by embolization was estimated by comparing the angiographic images taken at the beginning vs. the end of SAE. This comparison was performed retrospectively by an experienced interventional radiologist (R.L.) by first measuring the surface area on two-dimensional images and then calculating the percentage lost by extrapolating the results to the corresponding volume, in mL, as follows: percentage volume lost = 100 − ((100 × final volume)/initial volume). 

### 2.3. Outcome Measures 

Technical success was defined as complete exclusion of the iatrogenic renal arterial lesions visualized by angiography at the end of the procedure.

The primary outcome was the clinical success rate of SAE. Clinical success was defined as the absence of further embolization and/or surgery required to stop bleeding throughout the follow-up period.

The secondary outcomes were the change in creatinine clearance evaluated by the MDRD equation, the change in CKD stage, and the kidney volume lost due to SAE. We also compared outcomes across embolization materials used. 

### 2.4. Computed Tomography (CT) Acquisition

Of our 25 patients, 22 underwent CT as part of their preoperative etiological workup. All CT examinations were performed with a 320-detector row CT scanner (Aquilion One Genesis, Canon Medical Systems, Tokyo, Japan) or a dual energy Somatom Force machine (Siemens Healthineers, Erlangen, Germany). After an unenhanced acquisition, an iodinated contrast agent was injected, and acquisitions were obtained at the arterial phase (about 25 s postinjection) and at the portal phase (about 80 s postinjection) or nephrogenic phase (about 120 s postinjection). The optimal CT phase in this indication is the arterial phase, which visualizes FAAs as well-limited extravascular structures whose enhancement does not increase on later acquisitions. AVFs are seen as a direct communication between a renal artery and a renal vein or, more often, detected based on indirect signs including abnormally increased enhancement of a renal venous structure at the arterial phase. CT scan was waived in three patients because they went directly to diagnostic angiography based on US duplex findings.

### 2.5. Selective Arterial Embolization (SAE)

All patients in this study were treated at the diagnostic and therapeutic radiology department of the Dijon University Hospital. A 5-French (Fr) or 6-Fr sheath was introduced into a femoral artery, and the renal artery was then catheterized using a catheter with an appropriate angulation, most commonly a Simmons type I or II or a Cobra type II catheter. Selective angiography with acquisition in a single plane (Philips Allura FD 20, Best, The Netherlands) was performed to visualize the FAA or AVF and to detect any active bleeding. Superselective catheterization of the renal arteries was then accomplished using a 2.0-Fr to 2.7-Fr Progreat^™^ (Terumo, Tokyo, Japan) microcatheter. Once the target vessel was identified, the vascular anomaly was excluded using coils (Ruby^™^ Coil Complex Standard, Penumbra, Berlin, Germany; or Concerto Fibered Coils^™^, Medtronic, CA, USA) of appropriate size or a liquid embolization agent (Glubran^®^2, GEM, Viareggio, Italy; Onyx^®^, Medtronic, CA, USA; Gelita-spon^®^, Novimed, Dietikon, Switzerland; or Easyx^®^, Qmedics, Flurlingen, Switzerland).

### 2.6. Statistical Analysis

For descriptive data, mean or median depending on the normality of the distribution ± standard deviation (SD) was given. For categorical variables, absolute values and percentages (n (%)) were given.

Creatinine clearance values before and after PN, before and just after embolization, and before and 6 to 12 months after embolization were compared using Student’s t-test for paired samples; *p* values < 0.05 were taken to indicate statistically significant differences. 

The analysis was performed using the BiostaTGV website (http://biostatgv.sentiweb.fr (accessed on 1 June 2021)).

## 3. Results

### 3.1. Study Population and Surgical Data

Between June 2010 and June 2020, 25 patients were referred to our interventional radiology unit (Dijon University Hospital) for SAE of iatrogenic renal artery injuries (FAA and AVF) after PN. During this 10-year period, 298 patients underwent PN at our institution, including 12 (4%) who sustained iatrogenic arterial injuries. The remaining 13 study patients underwent PN at other healthcare institutions and were referred to us for treatment of the iatrogenic vascular complication.

Table 1 reports the main patient characteristics at the time of embolization. Table 2 reports the characteristics of the arterial lesions and endovascular procedures.

### 3.2. Endovascular Treatment

All 25 patients had at least one FAA and the number of FAAs per patient ranged from 1 to 5.

Technical success was achieved after first embolization at 96% (24 patients) and 100% after a second embolization procedure.

Clinical success was achieved after first embolization at 96% (24 patients) and 100% after the second procedure.

The patient who required two embolization attempts had the second one performed 48 h after the first, due to a vasospasm during the first procedure that prevented complete embolization. The second attempt was technically and clinically successful. 

Figure 1 and Figure 2 show examples of successful endovascular embolization of FAAs.

The time from the first CT image to the first image taken during the endovascular procedure was known for 18 of the 22 patients who underwent diagnostic CT; the mean value was 292 ± 275 min (4 h and 52 min).

No major complications such as hematoma, false aneurysm at the puncture site, or vascular dissection were observed after SAE. In a patient with an FAA fistulized to the urinary tract, moderate biological glue leakage in the urinary excretory cavities was reported, with no complications or symptoms during follow-up. No recurrence detected clinically or by CT was recorded during the mean follow-up of 27.1 ± 24.3 months.

One patient experienced a renal and perirenal infection after PN and required RN 3 days after the embolization procedure, which had been technically and clinically successful. Two patients had residual hematuria, prompting contrast-enhanced CT and, in one patient, angiography. No evidence of recurrence of the arterial lesion was seen in these imaging studies. The hematuria resolved spontaneously within 5 days in both patients and was attributed to the resorption of urinary tract clots, which were seen in both patients as hyperattenuating foci on the unenhanced CT images.

Figure 3 shows an example of hyperattenuating foci on the unenhanced CT images.

### 3.3. Kidney Function after Selective Arterial Embolization (SAE)

The mean difference between the creatinine clearance values before and after PN was −10.96 mL·min·1.73 m^2^ (95% confidence interval (95%CI) [−17.79 to −4.50]; *p* < 0.01). The CKD stage worsened in nine (36%) patients and improved in two (8%) patients.

The mean difference between the creatinine clearance values before and 1–7 days after endovascular embolization was −5.51 mL·min·1.73 m^2^ (95%CI [−10.79 to −0.30]; *p* = 0.039). The CKD stage worsened in five (25%) patients and improved in two (8%) patients.

The mean difference between the creatinine clearance values before and 6–12 months after SAE was computed in 24 patients, as one patient was on long-term dialysis after RN for an infection. The change in creatinine clearance was not significant in these 24 patients (mean difference, −1.78 mL·min·1.73 m^2^; 95%CI [−5.80 to +2.23]; *p* = 0.37). The CKD stage worsened in four (16%) patients and improved in two (8%) patients.

Table 3 reports the renal function outcomes after partial nephrectomy and after selective endovascular embolization.

### 3.4. Change in Renal Functional Volume 

The mean change in renal functional volume between the start and the end of the endovascular embolization procedure, as estimated by angiography, was −13.8 ± 15.1%.

Figure 4 shows an example of renal functional volume change between the start and the end of the endovascular procedure. Figure 5 shows another example of glue selective embolization of an FAA in a kidney compressed by surrounding hematoma.

## 4. Discussion

The main findings from our study were that SAE of FAAs and AVFs induced during PN was technically and clinically effective in all patients, nearly always after a single attempt; did not induce any complications; and was associated with only a small and nonsignificant decline in creatinine clearance measured after 6–12 months of follow-up. About one-third of patients were managed with coils, one-third with a liquid embolic agent, and one-third with both. FAAs were far more common than AVFs, and no patient had an AVF with no FAA. 

Among patients who underwent PN at the Dijon University Hospital during the 10-year study period, 4% experienced iatrogenic renal artery injuries. This proportion is consonant with the 1% to 4.3% range reported in the literature [19,20]. More than a quarter of our patients had both FAAs and AVFs. In a retrospective study of a prospective database, 5/15 (33%) had both abnormalities [20].

The mean time from PN to CT or angiography to look for vascular injuries was 14.8 ± 12.6 days (range, 1–55 days), in keeping with earlier reports of mean times varying between 12 and 14.5 days (range, 1–33 days) from PN to diagnostic imaging or symptom onset [12,19,21,22,23,24,25,26,27,28,29]. Similarly, the mean time from nephrolithotripsy to bleeding was 10.5 days in a multicenter retrospective study of 144 patients reported in 2015, suggesting a common pathophysiology to the vascular complications after the two procedures [30]. Most vascular injuries due to PN become symptomatic within the first month, with a sharp peak at 2 weeks. Within this timeframe, the development of suggestive manifestations (hematuria, anemia, surgical wound bleeding, or flank pain) requires a diagnostic imaging study. We believe that angiography should be the first-line imaging study (as performed in three of our patients), as this strategy decreases the time to treatment, radiation dose to the patient, and renal nephrotoxicity of contrast agent exposure (by avoiding the injection for enhanced CT). The risk of iodinated contrast medium-associated nephropathy increases with the amount of contrast agent injected [31,32]. However, CT angiography is more sensitive as it detects bleeding rates of 0.3 to 0.5 mL/min compared with 0.5 to 1 mL/min for angiography. For small FAAs and AVFs, CT angiography allows more accurate localization of the lesion and detects any anatomical variants or stenotic arterial segments before the angiography. However, we found no studies of the embolization success rates with and without prior CT angiography. 

In our study, all 25 patients had suggestive symptoms consisting of hematuria, anemia, bleeding through the surgical wound, or flank pain. Iatrogenic arterial lesions have been described by others as incidental findings [21,26,33,34]. A 2013 literature review estimated that 3% of cases were asymptomatic [11]. The low incidence of iatrogenic vascular complications and infrequency of asymptomatic cases support conventional CT cancer monitoring in patients who are asymptomatic after PN, although the optimal monitoring frequency is not agreed on [4]. With conventional CT monitoring, asymptomatic arterial injuries may go undetected and may resolve spontaneously before the first postoperative CT is performed. In a retrospective study of 20 patients with FAAs or AVFs manifesting as gross hematuria, two patients required no treatment and had spontaneous resolution of the abnormalities and two others had no abnormalities detected by angiography [12]. Although the high risk of rupture supports the treatment of all FAAs, no specific recommendation exists for the management of asymptomatic FAAs detected by CT with a negative angiography.

Reported technical and clinical embolization success rates for FAAs and AVFs after PN range from 80% to 100% but are usually above 90% [19,21,22,26,29,35,36,37,38]. In most studies, coils were the only embolic agents used [20,23,29,37]. We found no significant differences in success rates between coils, liquid embolization agents, and the combination of both. However, the high success rates and the small numbers of patients in each subgroup limited our ability to detect such differences. The endovascular procedures in our study were performed by different interventional radiologists with various levels of experience. Thus, endovascular embolization of arterial injuries induced by PN can probably be performed effectively and safely by most interventional radiologists.

The mean difference between creatinine clearance before and after PN was −10.96 mL·min·1.73 m^2^ (95%CI [−17.79 to −4.50]; *p* < 0.01), consistent with two other studies (−15.93 mL/min, *p* < 0.01 and −16.4 mL/min, *p* < 0.001, respectively) [19,20]. This decrease is related to the loss of nephrons inherent in the procedure. Creatinine clearance also decreased significantly early after embolization, but this difference was no longer apparent after a few months. This transient decrease in renal function is consistent with nephrotoxicity of the contrast agent used for CT (performed in 88% of our patients) and for angiography. 

Two other studies also found nonsignificant late variations of +0.08 mL/min and +1.7 mL/min [19,20]. A case–control study that matched patients who underwent embolization to patients without iatrogenic arterial lesions found that the estimated glomerular filtration rates (eGFRs) were similar in the two groups both before embolization and 6 months later [29]. Few patients in our study experienced CKD stage progression, in keeping with earlier data [19]. The preservation of the eGFR despite loss of kidney volume seems paradoxical, as a linear relationship between renal volume and eGFR before and after PN has been reported [39].

No significant risk factors for iatrogenic vascular lesions have been identified to date. Nevertheless, the risk may be higher when the tumor is located centrally and exhibits an endophytic growth pattern, as surgical excision is then more difficult, notably due to the proximity of the renal hilum [12,22]. Diabetes and chronic renal failure present before endovascular embolization have been suggested to increase the risk of renal failure after the procedure, although neither factor was statistically significant [19]. Furthermore, both conditions are known risk factors for iodinated contrast medium-associated nephropathy, which may therefore act as a confounding factor.

Our study has several limitations. First, the design was retrospective, and the sample size was small. Second, the loss of renal volume after embolization was estimated by an interventional radiologist at our department who may have underestimated the loss due to subjective bias. The estimated loss was smaller than reported previously in one study (mean loss of 25.2% ± 14.3% estimated based on CT in 10 patients [20]) but considerably greater than the median loss of 5% (range, 1%–50%) estimated based on angiography in another study [38]. Third, contrast-enhanced ultrasound was not used in our patients because it was not yet available on an emergency basis at our institution. Contrast-enhanced ultrasound combined with unenhanced CT may improve the evaluation of the retroperitoneal compartment and urinary tract while decreasing the use of iodinated contrast agents. Contrast-enhanced ultrasound has been used successfully in at least two published studies, in five and four patients, respectively [35,37]. 

## 5. Conclusions

Arterial injuries during PN are infrequent but can be life-threatening and require treatment as an emergency or relative emergency. Endovascular embolization, when available, has excellent technical and clinical success rates with few complications and should therefore be preferred over hemostatic surgery to preserve the nephron capital. Renal function was unchanged by the embolization procedure. 

Further studies are needed to identify risk factors for FAAs and AVFs after PN and to provide the basis for recommendations aimed at standardizing the treatment of these vascular complications, notably the choice of the embolic agent(s) according to each vascular abnormality configuration.

## Figures and Tables

**Figure 1 biomedicines-11-01935-f001:**
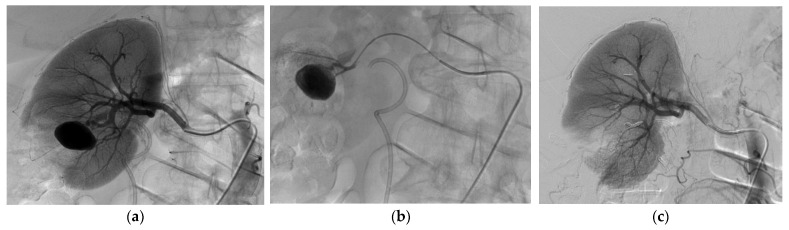
Endovascular treatment of an FAA after right partial nephrectomy. (**a**) Arteriogram showing the defect produced by partial nephrectomy and a single FAA near the surgical cut. (**b**) Arteriogram showing microcatheterization of the FAA. (**c**) Post-treatment control arteriogram showing complete exclusion of the renal FAA by the insertion of three coils to occlude the feeding renal artery.

**Figure 2 biomedicines-11-01935-f002:**
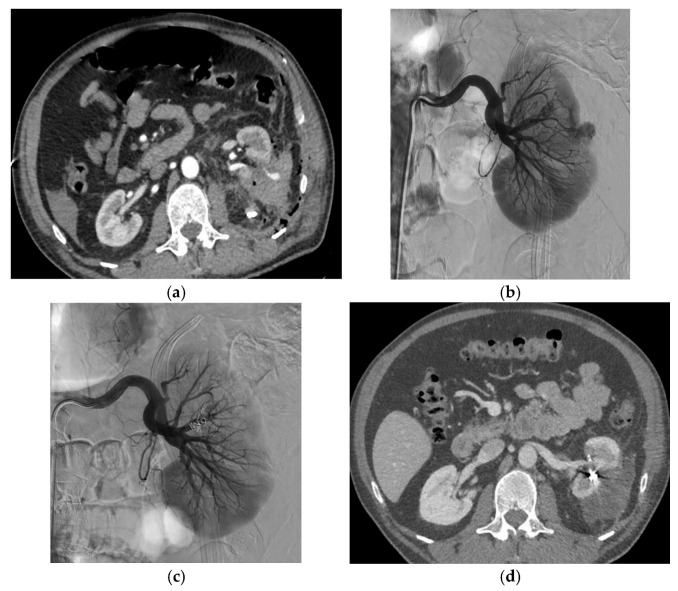
Endovascular embolization of an FAA after left partial nephrectomy. (**a**) Axial contrast-enhanced CT image at the arterial phase visualizing an FAA; a second FAA was seen on other images. (**b**) Left renal angiogram confirming the presence of two FAAs. (**c**) Renal angiogram showing complete exclusion of both FAAs after the insertion of three coils and the injection of a vial of the liquid embolization agent Easyx^™^ (iodinized polyvinyl alcohol (PVA) polymer ether, Antia Therapeutics AG, Bern, Switzerland). (**d**) CT scan after embolization showing the exclusion of FAAs.

**Figure 3 biomedicines-11-01935-f003:**
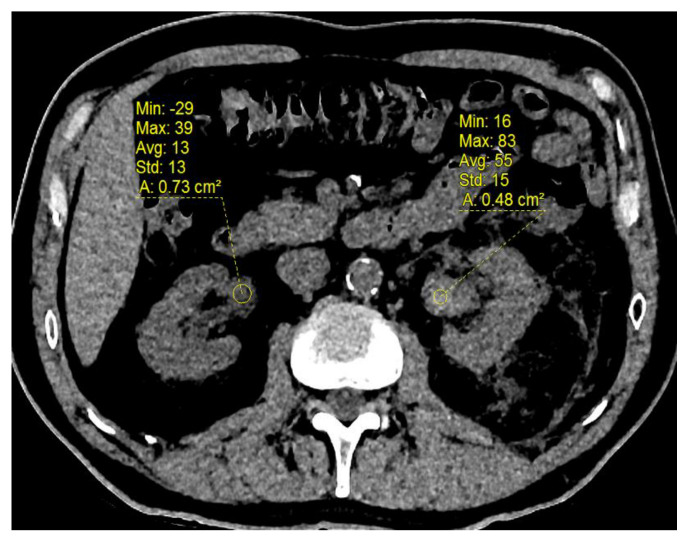
Unenhanced CT in a patient with residual hematuria after endovascular embolization of an FAA. A spontaneously hyperattenuating (+55 HU) image is visible in the left pyelocaliceal cavities, indicating presence of a blood clot. No bleeding or enhancement was found at the arterial or portal phase.

**Figure 4 biomedicines-11-01935-f004:**
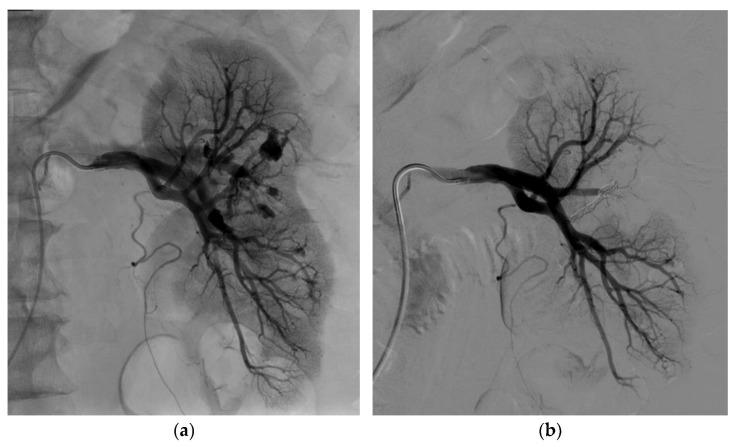
Change in renal volume between the start and end of the endovascular embolization procedure. (**a**) Left renal angiogram before embolization showing 3 FAAs and 2 AVFs. (**b**) Left renal angiogram after successful embolization by 3 micro-coils showing a 15% reduction in kidney volume as estimated based on the surface area on the two-dimensional image.

**Figure 5 biomedicines-11-01935-f005:**
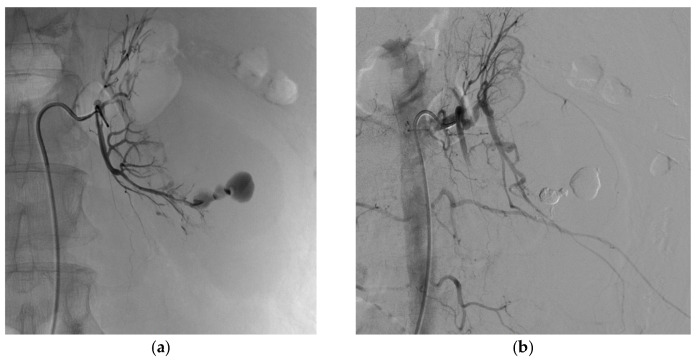
Endovascular glue selective embolization of an FAA after left partial nephrectomy. (**a**) Left renal angiogram showing the presence of an FAA. (**b**) Renal angiogram after glue embolization (Glubran ^®^2/Lipiodol^®^ in a 1:3 ratio) showing complete exclusion of the FAA.

**Table 1 biomedicines-11-01935-t001:** Mean features of the 25 study patients.

Variable	Mean ± SD or N (%)
Age (years)	66.6 ± 11.2
Males/Females	19 (76)/6 (24)
Body Mass Index (BMI) (kg/m^2^)	25.9 ± 4.3
Patients with BMI ≥ 30 kg/m^2^	4 (16)
Diabetes	2 (8)
Tumor size (cm)	3.9 ± 1.7
Surgical approach	
Lumbotomy and open PN	18 (72)
Robot-assisted transperitoneal PN	4 (16)
Laparoscopy completed by lumbotomy	3 (12)
Tumor pathology	
Clear-cell renal cell carcinoma	13 (52)
Oncocytoma	4 (16)
Papillary renal cell cancer	3 (12)
Chromophobe renal cell cancer	3 (12)
Cyst	3 (12)

y, year; N, number; %, percentage; SD, standard deviation; PN, partial nephrectomy.

**Table 2 biomedicines-11-01935-t002:** Characteristics of the arterial lesions and endovascular procedures in the 25 patients.

Variable	Mean ± SD or N (%)
Clinical presentation	
Gross hematuria	18 (72%)
Symptomatic anemia	9 (36%)
Flank pain	1 (4%)
Surgical wound bleeding	2 (8%)
Time from PN to diagnostic imaging (days)	14.8 (±12.6)
Iatrogenic vascular lesion	
Number of renal false arterial aneurysms	47
Number of renal arteriovenous fistulas	9
Number of patients with both	7 (28%)
Aneurysm size (mm)	10.5 (±4.7)
Diagnostic imaging	
Computed tomography ^a^	22 (88%)
Angiography	3 (12%)
Embolization technique	
Coils	8 (32%)
Liquid embolic agent	8 (6 Glubran2, 1 Onyx, 1 Gelatispon) (32%)
Coils and liquid embolic agent	9 (36%)
Outcomes	
Technical success at first attempt	24 (96%)
Technical success at first or second attempt	25 (100%)
Clinical success, one attempt	24 (96%)
Clinical success, one or two attempts	25 (100%)
Fluoroscopy time (min and s)	25 min 12 s (±19 min 35 s)
Radiation dose (mGy·cm^2^)	177,046 (±115,481)

^a^ unenhanced, then contrast-enhanced with arterial and portal or nephrogenic phases.

**Table 3 biomedicines-11-01935-t003:** Renal function outcomes after partial nephrectomy and after selective endovascular embolization.

Variable	Pre-Procedure	Post-Procedure	Difference	*p* Value
Partial nephrectomy				
eGFR, mean (±SD)	61.7 (±16.2)	50.6 (±14.5)	−10.96	<0.01
CKD stage, n of patients				
1	4	0		
2	14	14		
3	6	10		
4	1	0		
5	0	1		
Endovascular embolization		1–7 days		
eGFR, mean (±SD)	59.0 (±18.4)	50.2 (±19.5)	−5.51	0.039
CKD stage, n of patients				
1	1	1		
2	17	15		
3	6	6		
4	0	2		
5	1	1		
Endovascular embolization		6–12 months		
eGFR, mean (±SD)	58.5 (±14.6)	56.2 (±14.8)	−1.78	0.368
CKD stage, n of patients				
1	1	1		
2	17	15		
3	6	8		
4	0	0		
5	0	0		

eGFR: estimated glomerular filtration rate; CKD: chronic kidney disease.

## Data Availability

All the study data are reported in this article.

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
