# Peer review of "Selective Arterial Embolization of Pseudoaneurysms and Arteriovenous Fistulas after Partial Nephrectomy: Safety, Efficacy, and Mid-Term Outcomes"

_biomedicines, 2023, doi:10.3390/biomedicines11071935_

Round 1
Reviewer 1 Report
The authors have addressed the serious problem of renal surgery. In the introduction they have adequately highlighted the issue. The material and methods should be somewhat detailed. The number of patients is sufficient. The figures are of good quality and illustrate the work well. The literature is well selected.
[106] please specify the model of the CT scanner used
[107] please write why CT was waived in 3 patients
[116] whether the surgeries were carried out by different surgeons checking always the same
[118] change department to Department
Author Response
Responses to Reviewer 1 comments
The authors have addressed the serious problem of renal surgery. In the introduction they have adequately highlighted the issue. The material and methods should be somewhat detailed. The number of patients is sufficient. The figures are of good quality and illustrate the work well. The literature is well selected.
Q1 : [106] please specify the model of the CT scanner used
Reply : Thank you very much for your comment. It has been specified in the text as suggested. All CT examinations were performed with a 320-detector row CT scanner (Aquilion One Genesis, Canon Medical Systems) or a Dual Energy Somatom Force machine (Siemens Healthineers).
Q2 : [107] please write why CT was waived in 3 patients
Reply : Thank you very much for your comment. It has been written in the text in the last part of this section, as suggested. CT scan was waived in 3 patients because they went directly to diagnostic angiography based on US duplex findings.
Q3 : [116] whether the surgeries were carried out by different surgeons checking always the same
Reply : Thank you very much for your comment. Indeed, surgeries were carried out by different surgeons, but the small number of complications post-surgery did not allow us to identify any relationship between the operator and the risk of complications, by lack of power That’s why nothing has been added on this topic in the text.
Q4 : [118] change department to Department
Reply :Thank you very much for your comment. It has been changed as suggested.
Reviewer 2 Report
The authors presented an interesting article regarding selective arterial embolization of pseudoaneurysms and arteriovenous fistulas after partial nephrectomy. The techcniques employed were coils and embolization (and combined). Regarding the novelty, the techniques are known, but this representation is interesting as this is good success. The number of patients is small, but all limitations considering these techniques are noted in limitations. Statistical analysis is simple, but adequate for its purpose. Quaility of figures and tables is adequate. References are adequate.
Author Response
Responses to Reviewer 2 comments
The authors presented an interesting article regarding selective arterial embolization of pseudoaneurysms and arteriovenous fistulas after partial nephrectomy. The techniques employed were coils and embolization (and combined). Regarding the novelty, the techniques are known, but this representation is interesting as this is good success. The number of patients is small, but all limitations considering these techniques are noted in limitations. Statistical analysis is simple, but adequate for its purpose. Quality of figures and tables is adequate. References are adequate.
Reply : Thank you very much for your comments. Nothing new has been added, as requested.